# Families’ Worries during the First and Second COVID-19 Wave in Germany: Longitudinal Study in Two Population-Based Cohorts

**DOI:** 10.3390/ijerph19052820

**Published:** 2022-02-28

**Authors:** Susanne Brandstetter, Tanja Poulain, Mandy Vogel, Christof Meigen, Michael Melter, Angela Köninger, Christian Apfelbacher, Wieland Kiess, Michael Kabesch, Antje Körner

**Affiliations:** 1University Children’s Hospital Regensburg, Steinmetzstr. 1-3, 93049 Regensburg, Germany; michael.melter@ukr.de (M.M.); michael.kabesch@barmherzige-regensburg.de (M.K.); 2Member of the Research and Development Campus Regensburg (WECARE) at the Hospital St. Hedwig of the Order of St. John, Steinmetzstr. 1-3, 93049 Regensburg, Germany; angela.koeninger@barmherzige-regensburg.de (A.K.); christian.apfelbacher@med.ovgu.de (C.A.); 3LIFE Child, Leipzig Research Center for Civilization Diseases, Leipzig University, Philipp-Rosenthal-Strasse 27, 04103 Leipzig, Germany; tanja.poulain@medizin.uni-leipzig.de (T.P.); mandy.vogel@medizin.uni-leipzig.de (M.V.); christof.meigen@medizin.uni-leipzig.de (C.M.); wieland.kiess@medizin.uni-leipzig.de (W.K.); antje.koerner@medizin.uni-leipzig.de (A.K.); 4Department of Women and Child Health, University Hospital for Children and Adolescents, and Center for Pediatric Research, Leipzig University, Liebigstrasse 20a, 04103 Leipzig, Germany; 5University Department of Gynecology and Obstetrics, Hospital St. Hedwig of the Order of St. John, University Medical Center Regensburg, Steinmetzstr. 1-3, 93049 Regensburg, Germany; 6Institute of Social Medicine and Health Systems Research, Otto von Guericke University Magdeburg, Leipziger Str. 44, 39120 Magdeburg, Germany

**Keywords:** COVID-19, families, children, worries, regional differences

## Abstract

This study aimed to compare worries related to the Coronavirus disease 2019 (COVID-19) in families with young children in two regions in Germany differently affected by the pandemic (Regensburg in Southeast Germany, Leipzig in Eastern Germany) during the first and the second waves of the COVID-19 pandemic. 720 parents participating in the KUNO Kids health study in Regensburg (*n* = 507) or the LIFE Child study in Leipzig (*n* = 213) answered questions regarding COVID-19-related worries and trust in anti-pandemic policy measures during the first wave (spring 2020) and during the second wave (winter 2020/2021) of the pandemic. Ordinal mixed-effects models were performed to assess differences depending on region and time, adjusting for education and migration background. Participants worried most about the general economic situation and their family and least about their own health or financial situation. Worries about oneself, family, friends, hometown, and country were stronger during the second than during the first wave. In regional comparisons, worries about family, friends, and hometown increased more pronouncedly from wave 1 to wave 2 in Leipzig (OR ranging from 2.67 (95% CI 1.71–4.19) to 3.01 (95% CI 1.93–4.71), all *p* < 0.001) than in Regensburg (OR ranging from to 1.38 (95% CI 1.08–1.78) to 1.72 (95% CI 1.33–2.21), all *p* < 0.05), running parallel with the increase in SARS-CoV-2 infections. Trust in anti-pandemic policy measures, in contrast, decreased significantly between wave 1 and wave 2, with a stronger decrease in Regensburg (OR = 0.30 (95% CI 0.22–0.39), *p* < 0.001) than in Leipzig (OR = 0.91 (95% CI 0.59–1.41), n.s.). The degree of families’ COVID-19-related worries differs by region and time, which might be related to differences in infection rates and public interest. Regional differences should be taken into account when developing communication strategies and policy measures during the COVID-19 pandemic.

## 1. Introduction

Like most European countries, Germany was hit by the COVID-19 pandemic in different waves. The first wave (spring 2020) was characterized by unevenly distributed infection rates with large areas of only a few COVID-19 cases and some hotspot regions. The second and the third waves (winter 2020/21, spring 21) affected the whole of Germany, with especially high infection rates in East Germany. During all waves, lockdown measures were implemented, including comprehensive contact restrictions and long-term closures of daycare nurseries. 

Families were affected by the COVID-19 pandemic and the associated lockdown measures on several levels: they had to reorganize many aspects of everyday life, e.g., compensating restricted childcare, which had an impact on working conditions. Children themselves were confronted with the interruption of their familiar, daily routines and their social life [1], resulting in decreased health-related quality of life and more mental health problems [2,3,4]. Established support services for families were restricted, making access and utilization more difficult [5]. Previous studies conducted in Germany found increased stress levels in parents during the first COVID-19 wave compared to the period before the pandemic [6,7], as well as a greater burden or poorer well-being in parents than in non-parents [8,9]. Particularly in families with young children, parents showed a stronger decrease in well-being than non-parents [8]. As the COVID-19 pandemic has progressed, some people might have eventually adapted to the novel situation, and living with safety measures might have become the new normal. On the other hand, the continuous efforts necessary to maintain family life might be considered increasingly exhaustive the longer the COVID-19 pandemic has lasted, causing worries, stress and compromised well-being. 

Our study aimed to investigate families’ worries regarding different aspects of the COVID-19 pandemic. Moderate levels of worries can be beneficial as they were shown to be associated with favorable changes in safety and infection behaviors [10]. COVID-19 related worries can relate to the immediate health impact of a SARS-CoV-2 infection for oneself or others, but also to societal consequences of lockdown and safety measures [11]. Revealing what families with young children worry about can help in understanding the challenges of their current everyday living and in uncovering unmet support needs. Moreover, the extent of worries can be considered indicative for well-being. Continuous worrying compromises mental health and is closely linked to anxiety and depression. 

In our study, we were particularly interested in differences in families’ worries between the first and the second waves of the pandemic and in regional differences. By using data from two actively recruiting childhood cohorts (the KUNO Kids health study [12] and the LIFE Child study [13]) in the cities of Regensburg and Leipzig, located in the Southeast (Regensburg) and East (Leipzig) of Germany, we could shed light on the course of families’ worries in two regions with considerably different COVID-19 incidences. 

## 2. Materials and Methods

### 2.1. Design

The study adopted a longitudinal design. Data were collected through online surveys from two childhood cohorts in Germany. The cohorts are situated in Regensburg (Bavaria, Southeast of Germany) and Leipzig (Saxony, East of Germany). Compared to other German regions, Regensburg had high incidence rates during the first wave (7-day incidence (infections/100.000 inhabitants/7 days) = 11 (15.05.2020)) but was only moderately affected during the second wave (7-day incidence = 34 (01.02.2021)) [14]. Leipzig, in contrast, was only mildly affected by the COVID-19 pandemic during the first wave (7-day incidence = 1 (15.05.2020)) but was hit hard during the second wave (7-day incidence = 193 (01.02.2021)) [15]. Figure 1 summarizes the time trends of the COVID-19 incidences from March 2020 to April 2021 for both study regions. 

### 2.2. Study Participants

The KUNO-Kids health study is an ongoing, multi-purpose birth cohort, which started in 2015 and has had 3249 participants so far [12]. The study is located in Southeast Germany and covers the city of Regensburg (approximately 170.000 inhabitants) and the adjacent, mostly rural regions. All adult mothers with a basic understanding of German who give birth at the clinic St. Hedwig are asked to provide written informed consent for participation in the study. Only one child per family is included in the study. Data are collected immediately after the birth of the child and at various follow-ups. 

The LIFE Child study is an ongoing childhood cohort study that started in 2011 and has 4800 participants so far [13]. The study is conducted in Leipzig, a city with approximately 600.000 inhabitants situated in Saxony (Eastern Germany). Participants in LIFE Child are recruited from the prenatal period until the age of 16 years and participate in annual follow-up visits. All parents provide written informed consent before participation. 

All families with children aged 1.5 to 5.9 years, who were currently participating in either KUNO-Kids or LIFE Child and who had agreed to be contacted for additional studies, were eligible and contacted for the present study. In Regensburg, the invitation was sent to 1296 eligible families. Of these, 612 (50.1%) completed the first questionnaire (wave 1) between 7 May and 28 May 2020, and 507 (82.8% of those participating during the first wave) additionally completed the second questionnaire (second wave) between 16 January and 10 February 2021. In Leipzig, 306 out of 721 eligible families (42.4%) responded during the first wave between 23 April and 9 May 2020, and 213 (67.4%) of those completed the second questionnaire (second wave) between 18 January and 1 February 2021. 

The KUNO-Kids health study and the LIFE Child study were designed in accordance with the Declaration of Helsinki. The KUNO-Kids study was approved by the Ethics Committee of the University of Regensburg (Reference Number 14-101-0347). The LIFE Child study was approved by the Ethics Committee of the Medical Faculty of the Leipzig University (Reg. No. 264/10-ek). All participants were informed on the study content and provided informed written consent before participation in the KUNO-Kids study or the LIFE Child study.

### 2.3. Questionnaire

The survey covered the same questions at both time points. Topics included SARS-CoV-2 infections among family members or friends, risk persons among family and friends, current quarantine status, current working situation, trust in anti-pandemic policy measures and various COVID-19-related worries (see Table A1). For both cohorts, the survey data were complemented by information on the age of mother and child, migration background (at least one parent born in a country other than Germany versus both parents born in Germany), highest educational level (university entrance level of at least one parent versus no university entrance level), the employment status of both parents (yes versus no) and the child’s nursery or kindergarten attendance.

### 2.4. Statistical Analyses

All analyses were performed using R version 4.0 [16]. Categorical and ordinal data are reported as frequencies and percentages, continuous data as means and ranges. Associations between COVID-19-related worries, trust in anti-pandemic policy measures (ordered ordinal scaled outcomes), time (wave 2 versus wave 1) and region (Leipzig versus Regensburg), as well as interactions between time and region, were analyzed using ordinal mixed-effects models [17], with the subject included as a random effect. Effects were presented as odds ratios (ORs) together with 95% confidence intervals (CIs). Interactions between time and region were included if the interaction term reached statistical significance (*p* < 0.05). In the case of significant interactions between time and region, the main effects should not be interpreted and, therefore, were not presented. All analyses were adjusted for education and migration background.

## 3. Results

Sociodemographic and pandemic-related information on the study participants is displayed in Table 1. The sample was characterized by a high socio-economic status, reflected in high rates of employment (employment of both parents: 57% (Leipzig)–62% (Regensburg)), high rates of families with at least one parent with university entrance level (79% (Regensburg)–85% (Leipzig)) and a low rate of migration background (3% (Leipzig)–10% (Regensburg)). The percentage of persons in quarantine (1–2%) was negligible in both cities and at both time points. The majority of participants (>84%) knew at least one person at risk. This percentage was highest at wave 2 in Regensburg (96%). The percentages of participants knowing persons infected with COVID-19 increased dramatically from wave 1 (20%) to wave 2 (67%). At wave 1, this percentage was higher in Regensburg (23%) than in Leipzig (13%). At wave 2, in contrast, the percentage was slightly higher in Leipzig (70%) than in Regensburg (65%). In both regions, most families (about 60%) stated that at least one parent worked from home more often than before the pandemic.

The percentage of participants stating strong to extreme COVID-19-related worries and trust in the anti-pandemic policy measures in Regensburg and Leipzig at wave 1 and wave 2 are shown in Figure 2. In both regions, participants worried most about the economy, with more than 60% stating that they worry very much or extremely. Worries about themselves or their own financial situation were reported least frequently, with less than 20% stating that they worry very much or extremely. Trust in anti-pandemic policy measures was high, especially in Regensburg (see Figure 2). 

Table 2 presents the effects of time and region on COVID-19-related worries and trust in anti-pandemic policy measures. While worries about the world (OR = 0.80 (95% CI 0.65–0.99), *p* < 0.05), the economy (OR = 0.74 (95% CI 0.60–0.92), *p* < 0.01), and one’s own financial situation (OR = 0.54 (95% CI 0.43–0.68), *p* < 0.001) decreased significantly from wave 1 to wave 2, worries about oneself (OR = 2.08 (95% CI 1.66–2.60), *p* < 0.001) and one’s own country (OR = 1.54 (95% CI 1.25–1.90), *p* < 0.001) increased. Worries about family, friends, and hometown also increased significantly, with a significantly higher increase in Leipzig than in Regensburg. For example, the OR for the increase of worries about the family was 1.38 (95% CI 1.08–1.78, *p* < 0.05) in Regensburg and 3.01 (95% CI 1.93–4.71, *p* < 0.001) in Leipzig. 

Regarding further differences between Leipzig and Regensburg, the participants worried significantly less about themselves (OR = 0.47 (95% CI 0.31–0.72), *p* < 0.001) and the world (OR = 0.64 (95% CI 0.43–0.95), *p* < 0.05) in Leipzig than in Regensburg. With respect to trust in anti-pandemic policy measures, we observed a significant decrease from wave 1 to wave 2 in Regensburg (OR = 0.30 (95% CI 0.22–0.39), *p* < 0.001), but no significant change in Leipzig (OR = 0.91 (95% CI 0.59–1.41), *p* = 0.689), where it was already lower in the first wave (see Figure 2).

## 4. Discussion

The present longitudinal online study investigated COVID-19-related worries in families with young children during the first (spring 2020) and second (winter 2020/2021) waves in two regions in Germany (Regensburg and Leipzig) that have been affected differently by the pandemic. The online surveys were completed by parents of 1.5- to 5-year-old children, i.e., by individuals who often suffered from a double burden during the pandemic (work and caring for children at home) [18,19]. Indeed, 60% of our study sample reported switching to a home office because of the pandemic.

Overall, study participants worried most about the possible impact of the pandemic on the economy. They also worried about the situation in the world or their country. These worries might be explained by the economic crises caused by the COVID-19 pandemic [20,21] and the omnipresence of the economic consequences related to the pandemic (through media reports, closed businesses, knowledge of acquaintances who lost their job or were on short-time work). In contrast, the worries about their own financial situation were comparably low. One reason for this discrepancy might be the low proportion of people with a migration background and the large proportion of people with higher education, indicating the high socio-economic status of the study participants. In higher social strata, people are more likely to have stable jobs and financial reserves and, therefore, suffer less from the pandemic-related economic crisis [22]. Other studies found that, in particular, socially disadvantaged families were affected by lockdown measures [23,24].

Regarding worries about themselves and close relatives, study participants worried most about family members and least about themselves. Worries about family members might particularly reflect worries about (grand)parents, as older people are at higher risk of severe or even fatal illness relating to COVID-19 than younger people [25]. The low level of worries about themselves might be explained by the higher social status in the present sample, as people from higher social classes are at lower risk regarding health risks at work [22]. The finding might also reflect a perceived resilience to infection or a strong belief in a quick recovery if infected, e.g., because one does not belong to a risk group [26] or follows a healthy lifestyle [27]. 

Our analyses showed that most COVID-19-related worries assessed in the present study increased from the first to the second wave. This increase was particularly strong for worries referring to participants themselves or to their immediate vicinity (family, friends, hometown). In the spring of 2020, many people still assumed that the virus would not affect their lives in the long term. By the winter of 2020/2021, however, it had become clear that the virus had spread further, with considerable mortality. The 7-day incidence was also higher during the second wave (34/100.000 and 193/100.000 inhabitants/week in Regensburg and Leipzig, respectively [14,15]) than during the first wave (11/100.000 and 1/100.000 inhabitants/week), as was the number of infected people in the circle of acquaintances (70% and 65% during wave 2 versus 23% and 13% during wave 1) in the present sample. Therefore, the increase in worries from the first to the second wave might reflect the increase in actual infections and the higher visibility of the pandemic. This assumption is in line with findings from other studies. A survey among families in Australia with 14 repeated assessments showed that infection rates and mental health indicators over time corresponded to each other [28]. An Austrian study showed lower wellbeing and higher stress levels during the first COVID-19-related lockdown, when infection rates were high, than six months later, when infection rates were low [29]. A study conducted in Germany also found a significant decline in wellbeing from the first to the second wave [30]. The authors also observed a decrease in safety behavior and, therefore, interpreted the findings as pandemic fatigue (rather than increased own concern).

Interestingly, for some worries, namely worries about family members, friends and hometown, the increase from the first to the second wave was stronger in Leipzig than in Regensburg. This finding might be explained by the higher increase in the number of infections in Leipzig. 

Taken together, the variations in worries between different time points or different study regions might be explained by (the development of) the regional incidence of infections and, related to this, by one’s confrontation with the virus or the pandemic. This is in line with findings from a Germany-wide study that revealed regional differences in people’s mental health according to infection rates [31].

For worries about oneself and worries about the world, we observed a constantly lower level of worries in Leipzig than in Regensburg, which is hardly explainable by the course of the pandemic only. General differences between the two regions could play a role here; these can result from the aftermath of the different socio-political systems in East and West Germany but also from current differences in socio-economic living conditions, and cannot easily be interpreted. 

Regarding trust in anti-pandemic policy measures, our analyses showed a moderately high level of trust, which, however, was significantly lower during the second than during the first wave. These findings are in line with studies conducted in Germany [32,33] and Austria [34], according to which people’s acceptance of measures and trust in the scientific basis of measures decreased significantly from summer 2020 to winter 2021.

Our study has some implications for public health research and practice: Although many study participants expressed trust in political measures at the beginning, this proportion declined during the course of the pandemic. The decrease in trust observed in our study might reflect families’ dissatisfaction or lack of understanding in the face of constantly changing restriction measures and relaxations. For example, the rules for closure or opening of nurseries have changed continuously, impeding families’ organization of everyday life. The public debate about adequate safety measures in childcare facilities was marked by controversies, and the actual implementation of safety measures differed between different regions of Germany. However, it is essential for the management of the pandemic that people understand why measures are implemented. Policy measures should be comprehensible and consistent, and better communication about why a specific measure is implemented could eventually contribute to people’s acceptance and trust. A further implication can be derived from our finding of variability of worries and trust over time and between regions. People in severely affected regions expressed stronger concerns, but they might also be more prone to implement safety measures. Policy measures and the associated communication strategy should acknowledge regional differences and adapt to the local situation.

### Strengths and Limitations

We have learned during the COVID-19 pandemic that infection rates and associated anti-pandemic policy measures are rapidly changing and can vary greatly from region to region. Our study acknowledges this variability by investigating large samples from two regions in Germany at two time points and allows for a close look at parents’ worries during the COVID-19 pandemic. By using online surveys, families could be contacted in a timely fashion (immediately after the first and the second COVID-19 wave) and the effort for participants was low. 

However, we could not control in which context participants completed the questions. A limitation of our study is that the study population consisted mostly of well-educated parents, as we drew them from our ongoing childhood cohort studies. As with most studies relying on active participation [35,36], families with low education, low income and migration background were underrepresented. Due to this selection bias, the generalizability of our findings is limited. People who were more concerned about the pandemic or who considered the pandemic a topic relevant to be investigated might have been more prone to participate in the present surveys. On the other hand, we may underestimate the impact of the pandemic on families who are underprivileged, and who are more severely affected by the pandemic, e.g., because their children attend child care centers with higher infection rates more frequently [37] or because they are single parents and, therefore, experience more care-related worries [38]. Further, although we are studying two time points and two study regions, we cannot differentiate between the effects caused by differences in the course of the pandemic, by the associated policy measures and by differences of the two regions. Additionally, residual confounding cannot be excluded. Additional limitations concern the selection of questions for the survey. Some important pandemic-related worries, such as fear of losing one’s job, were not captured. Regarding worries about oneself, family, friends, hometown, country and the world, it is not clear whether the worries are related to COVID-19 or to anti-pandemic measures. Finally, we could not use an established validated measurement instrument; instead, the questionnaire was developed by the authors. Therefore, comparisons with other studies are only possible to a very limited extent. 

## 5. Conclusions

Investigating two large samples of predominantly well-situated families in Germany, we found that COVID-19-related worries and trust in anti-pandemic policy measures changed during the course of the pandemic and differed remarkably between the study regions. The extent to which worries or trust were expressed might depend on regional infection rates and the associated salience of the pandemic, as well as the anti-pandemic policy measures. Acknowledging regional differences and implementing safety measures adapted to the current local situation might help families to accept and eventually to better cope with the burden of the COVID-19 pandemic.

## Figures and Tables

**Figure 1 ijerph-19-02820-f001:**
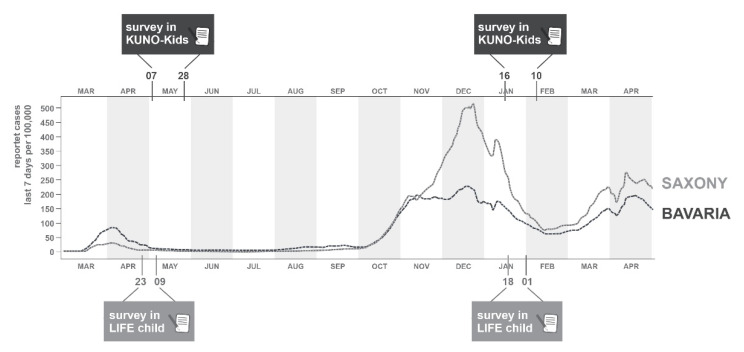
Course of COVID-19 incidence in Bavaria (for Regensburg) and Saxony (for Leipzig) and timing of online surveys.

**Figure 2 ijerph-19-02820-f002:**
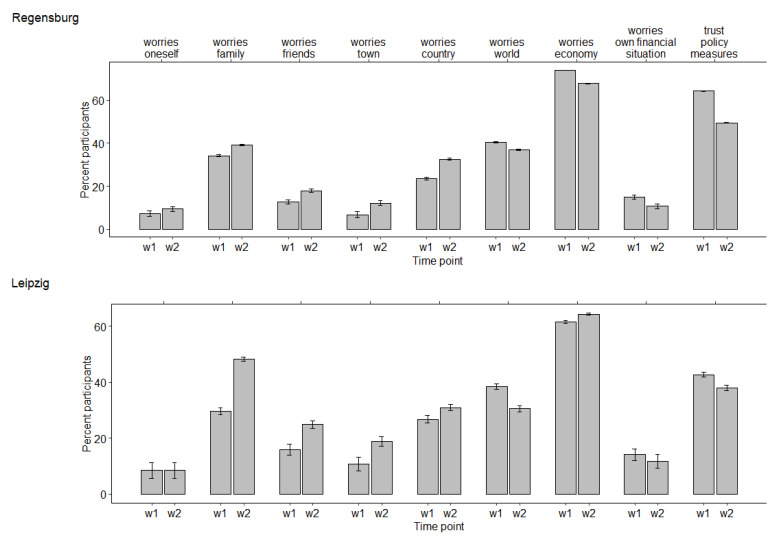
Percentage (+95% CI) of COVID-19 related worries and trust in policy measures in Regensburg and Leipzig at wave 1 (w1) and wave 2 (w2). For clarity, only the percentages of participants indicating that they worry (or trust) “very much” or “extremely” are presented.

**Table 1 ijerph-19-02820-t001:** Characteristics of the study populations in Regensburg and Leipzig at wave 1 and wave 2 of the COVID-19 pandemic.

		Regensburg (*n* = 507)		Leipzig (*n* = 213)	
		w 1	w 2	w 1	w 2
Sociodemographic information					
Age mother at wave 1	Mean (SD)	36.4 years (3.9 years)		35.5 years (4.5 years)	
Age child at wave 1 ^a^	Mean (SD)	3.4 years (0.9 years)		3.7 years (1.3 years)	
Child in kindergarten ^a^	Yes	334 (66%)		133 (62%)	
	No	148 (29%)		38 (18%)	
	Missing	25 (5%)		42 (20%)	
Employment status parents	Both employed	316 (62%)		122 (57%)	
	One employed	159 (31%)		66 (31%)	
	Both unemployed	4 (1%)		7 (3%)	
	Missing	28 (6%)		18 (9%)	
Education ^b^	Higher	400 (79%)		180 (85%)	
	Lower	105 (21%)		32 (15%)	
	Missing	2 (<1%)		1 (<1%)	
Migration background ^c^	Yes	53 (10%)		7 (3%)	
	No	441 (87%)		154 (72%)	
	Missing	13 (3%)		52 (25%)	
Pandemic-related information					
Currently in quarantine	Yes	2 (<1%)	6 (1%)	5 (2%)	5 (2%)
	No	505 (100%)	501 (99%)	208 (98%)	208 (98%)
Person at risk in family/circle of friends	Yes	429 (85%)	488 (96%)	179 (84%)	184 (86%)
	No	78 (15%)	19 (4%)	34 (16%)	29 (14%)
Infection in family/circle of friends	Severe	49 (9%)	102 (20%)	10 (5%)	46 (22%)
	Mild	69 (14%)	230 (45%)	18 (8%)	102 (48%)
	No	389 (77%)	175 (35%)	185 (87%)	65 (30%)
Change in working situation	No change	185 (36%)		85 (40%)	
	More home office	304 (60%)		128 (60%)	
	Missing	18 (4%)		0 (0%)	

w = wave; SD = standard deviation; ^a^ Only the child participating in the KUNO-Kids or LIFE Child study was considered; ^b^ higher: university entrance level of mother or father; lower: no university entrance level of mother and father; ^c^ defined as one or both parents born in a country other than Germany.

**Table 2 ijerph-19-02820-t002:** Effects of time (wave 1 versus wave 2) and region (Leipzig versus Regensburg) on COVID-19-related worries and trust in policy measures.

		Main Effects Only		Significant Interaction Time * Region		
		Effect Region (L vs. RB)	Effect Time (w 2 vs. w 1)	Effect Time in Regensburg	Effect Time in Leipzig	*p*-Value Difference
Worries about own person	OR CI	0.47 0.31–0.72	2.08 1.66–2.60	ns		
	*p*	<0.001	<0.001			
Worries about family	OR CI			1.38 1.08–1.78	3.01 1.93–4.71	
	*p*			<0.05	<0.001	<0.001
Worries about friends	OR CI			1.72 1.33–2.21	2.97 1.90–4.67	
	*p*			<0.001	<0.001	<0.01
Worries about hometown	OR CI			1.47 1.15–1.88	2.67 1.71–4.19	
	*p*			<0.01	<0.001	<0.05
Worries about country	OR CI	0.76 0.54–1.06	1.54 1.25–1.90	ns		
	*p*	0.108	<0.001			
Worries about world	OR CI	0.64 0.43–0.95	0.80 0.65–0.99	ns		
	*p*	<0.05	<0.05			
Worries about economy	OR CI	0.76 0.52–1.11	0.74 0.60–0.92	ns		
	*p*	0.159	<0.01			
Worries about own financial situation	OR CI	0.82 0.49–1.35	0.54 0.43–0.68	ns		
	*p*	0.430	<0.001			
Trust in policy measures	OR CI			0.30 0.22–0.39	0.91 0.59–1.41	
	*p*			<0.001	0.689	<0.001

Interaction terms are only presented in the case of significance (*p* < 0.05). In the case of significant interactions between Time ***** Region, main effects should not be interpreted and, therefore, are not presented. All associations are adjusted for education and migration. w = wave; ns = not significant; OR = odds ratio; CI = 95% confidence interval.

## Data Availability

De-identified participant data of the KUNO Kids study will be made available upon reasonable request from MK. Data of the LIFE Child study cannot be shared publicly because the data contains potentially sensitive information and publishing data sets is not covered by the informed consent provided by the study participants. Researchers interested in accessing and analyzing data collected in the LIFE Child study may contact the data use and access committee (forschungsdaten@medizin.uni-leipzig.de).

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
