# Peer review of "Families’ Worries during the First and Second COVID-19 Wave in Germany: Longitudinal Study in Two Population-Based Cohorts"

_ijerph, 2022, doi:10.3390/ijerph19052820_

Round 1

Reviewer 1 Report

The manuscript reports results of a longitudinal study in two populational childhood cohorts on families` worries during the first and second COVID-19 wave 2 in Germany. The topic is interesting, however, there are several shortcomings in comprehensibility and readability of the paper. Although the study design would allow for more robust explanations and interpretations, hence the number of participants is relatively low for a large country like Germany, the results are only narratively reported and in-depth practical and theoretical implications of the result are unfortunately lacking.

Some comments:

The title is very long and could be shortened. Population or population-based is a more common term.

The abbreviation of authors is missing in the affiliation part.

The abstract should be more precise and also offering figures to picture decrease or increase and a stronger conclusion.

The introduction is very short and does not reflect the magnitude on studies on this topic. Also, the term worries is not introduced.

What is the rationale behind the study?

In the table, it is unclear that the total numbers refer to parents. If a family has more than one child, how is the mean age etc. calculated (maybe I have read over this information)? Age should be presented as means and SD.

How is migration background defined?

Fig 2 could profit from error bars.

Please check for formatting issues.

As statistical analyses were performed, the results should be adequately presented in the text. The magnitude of decreased significantly etc. is unclear.

Discussion:

The term financial crisis is misleading, as it refers to past events.

Please explain potential general differences in mentality.

The limitations section should also address problems of survey in general and aspects of generalizability.

Practical and theoretical implications of the result should be presented in detail.

Author Response

RESPONSE

We would like to thank the reviewers and the editors for their suggestions and constructive feedback to our manuscript. Based on these, we have prepared a revised version of our manuscript.

Reviewer 1:

The manuscript reports results of a longitudinal study in two populational childhood cohorts on families` worries during the first and second COVID-19 wave 2 in Germany. The topic is interesting, however, there are several shortcomings in comprehensibility and readability of the paper. Although the study design would allow for more robust explanations and interpretations, hence the number of participants is relatively low for a large country like Germany, the results are only narratively reported and in-depth practical and theoretical implications of the result are unfortunately lacking.
Response: Thank you for your helpful comments to our manuscript. The introduction and the discussion section have been expanded in order to allow for better comprehensibility and more thorough discussion of our findings and potential implications. We have prepared a revised version and address your comments as follows.

Some comments:
The title is very long and could be shortened. Population or population-based is a more common term.
Response: We shortened the title as follows: “Families` worries during the first and second COVID-19 wave in Germany: longitudinal study in two population-based cohorts”

The abbreviation of authors is missing in the affiliation part.
Response: We included authors’ email addresses and abbreviations in the affiliation part.

The abstract should be more precise and also offering figures to picture decrease or increase and a stronger conclusion.
Response: Thank you for this suggestion. We revised the abstract according to your suggestions in order to make it more precise. We now provide the ORs for specific associations. We also state clearly which COVID-19-related worries were stronger during the second than during the first wave. In addition, we included a stronger conclusion.

Changes in the manuscript:
Abstract – Results and Conclusion: “Participants worried most about the general economic situation and their family and least about their own health or financial situation. Worries about oneself, family, friends, hometown, and country were stronger during the second than during the first wave. In regional comparisons, worries about family, friends, and hometown increased more pronouncedly from wave 1 to wave 2 in Leipzig (OR ranging from 2.67 – 3.01, all p < .001) than in Regensburg (OR ranging from to 1.38 to 1.72, all p < .05), paralleling the increase in SARS-CoV-2 infections. Trust in anti-pandemic policy measures, in contrast, decreased significantly between wave 1 and wave 2, with a stronger decrease in Regensburg (OR = 0.30, p < .001) than in Leipzig (OR = 0.91, n.s.). The degree of families’ COVID-19-related worries differs by region and time, which might be related to differences in infection rates and public interest. Regional differences should be taken into account when developing communication strategies and policy measures during the COVID-19 pandemic.”

The introduction is very short and does not reflect the magnitude on studies on this topic. Also, the term worries is not introduced.
Response: We agree with the reviewer that there is now a large and steadily growing body of research on families in the COVID-19 pandemic. Therefore, we have expanded the introduction section and have included more current studies. We also focus more on the concept of worries and explain why we chose to investigate parental worries.

Changes in the manuscript:
Introduction:  “Families were affected by the COVID-19 pandemic and the associated lockdown measures on several levels: they had to reorganize many aspects of everyday life, e.g., compensating restricted childcare, which had an impact on working conditions. Children themselves were confronted with the interruption of their familiar, daily routines and their social life [1], resulting in decreased health-related quality of life and more mental health problems [2–4]. Established support services for families were restricted making access and utiliza-tion more difficult [5].Previous studies conducted in Germany found increased stress lev-els in parents during the first COVID-19 wave compared to the period before the pandemic [6,7] as well as more burden or poorer well-being in parents than in non-parents [8,9]. Par-ticularly in families with young children, parents showed a stronger decrease in well-being than non-parents [8]. As the COVID-19 pandemic has progressed and some people might have eventually adapted to the novel situation and living with safety measures might have become the new normal. On the other hand, the continuous efforts necessary to maintain family life might be considered increasingly exhaustive the longer the COVID-19 pandemic has lasted, causing worries, stress and compromised well-being.
Our study aimed to investigate families’ worries regarding different aspects of the COVID-19 pandemic. Moderate levels of worries can be beneficial as they were shown to be associated with favorable changes in safety and infection behaviours [10]. COVID-19 related worries can relate to the immediate health impact of a SARS-CoV-2 infection for oneself or others but also to societal consequences of lockdown and safety measures [11]. Revealing what families with young children worry about can help understanding the challenges of their current everyday living and uncovering unmet support needs. Moreo-ver, the extent of worries can be considered indicative for well-being. Continuous worrying compromises mental health and is closely linked to anxiety and depression.”

What is the rationale behind the study?
Response: The rationale behind our study was to investigate parental worries regarding the COVID-19 pandemic, with a special focus on how worries changed during the course of the pandemic and between different regions. With a better understanding of the prevalence of different worries and its development we can shed light on the situation of families with young children and eventually contribute to the development of supportive measures if necessary. We believe that by expanding the introduction of our manuscript the rationale behind our study has now become clearer.

In the table, it is unclear that the total numbers refer to parents. If a family has more than one child, how is the mean age etc. calculated (maybe I have read over this information)? Age should be presented as means and SD.
Response:
In the Table caption, we added the notion that information on children (age and kindergarten) only refers to the child participating in the KUNO-Kids or LIFE Child study. We added the standard deviations for age of mother and child.

Changes in the manuscript: see Table 1

How is migration background defined?
Response:
Migration background was defined as one parent being born in a country other than Germany. This information was already included in the table caption. Now, we provide this definition additionally in the method section. We also included definitions of how the different categories of employment status and education were built.

Changes in the manuscript:
Methods: “For both cohorts, the survey data were complemented by information on the age of mother and child, migration background (at least one parent born in a country other than Ger-many versus both parents born in Germany), highest educational level (university en-trance level of at least one parent versus no university entrance level) and employment status of both parents(yes versus no), and the child’s nursery or kindergarten attendance.”

Fig 2 could profit from error bars.
Response:
We revised Figure 2 and included 95% confidence intervals for the prevalences.

Changes in the manuscript: see Figure 2

Please check for formatting issues.
Response: We checked for formatting issues and corrected them throughout the manuscript.

As statistical analyses were performed, the results should be adequately presented in the text. The magnitude of decreased significantly etc. is unclear.
Response:
We thank the reviewer for this helpful comment. In the previous version of the manuscript most of the statistical findings were presented mainly in the tables. We have now included more information on the direction and size of effects (ORs) and their statistical significance (CIs, p values) in the result section.

Changes in the manuscript: see Results

The term financial crisis is misleading, as it refers to past events.
Response:
Thank you for this comment. We now use the term “economic crisis” throughout the discussion.

Please explain potential general differences in mentality.
Response: We found differences in participants’ answers between the two study regions which could not be explained by the distribution of the two samples or by the different course of the pandemic. When seeking for possible reasons for these differences it is obvious to consider differences in mentality and attitudes between the two study regions situated in East and West Germany. But even though there are still differences 30 years after the German reunification in specific attitudes towards politics or family life (IFO- Institute, 2017) and in some health indicators (Lampert et. al, 2019, Journal of Health Monitoring) these differences cannot be easily attributed and interpreted. Many researchers suggested to investigate regional differences on a smaller scale. Taken together, we now refer to potential differences between East and West Germany when discussing the findings of our study, but we also suggest that this explanation might fall short.

Changes in the manuscript:
Discussion: “General differences  between the two regions could play a role here these can result from the aftermath of the different socio-political systems in East and West Germany but also from current differences in socio-economic living conditions, and cannot easily be interpreted.”

The limitations section should also address problems of survey in general and aspects of generalizability.
Response:
We now have expanded the section on limitations and refer to problems of online surveys and to selection bias/ lack of generalizability.

Changes in the manuscript:
Discussion: “By using online surveys families could be contacted timely (immediately after the first and the second COVID-19 wave) and the effort for participants was low. However, we could not control in which context participants completed the questions.”

“A limitation of our study is that the study population consisted mostly of well-educated parents as we drew them from our ongoing childhood cohort studies. As with most studies relying on active participation [35,36], families with low education, low income, and migration background were underrepresented. Due to this selection bias generalizability of our findings is limited.  People who were more concerned about the pandemic or who considered the pandemic a topic relevant to be investigated might have been more prone to participate in the present surveys. On the other hand, we may underestimate the impact of the pandemic on families who are underprivileged, and who are more severely affected by the pandemic, e.g., because their children attend more frequently child care centers with higher infection rates[37] or because they are single parents and, therefore, experience more care-related worries [38].”

Practical and theoretical implications of the result should be presented in detail.
Response: We now provide a thorough discussion of implications for public health research and practice which can be drawn from our study.

Changes in the manuscript:
Discussion: “Our study has some implications for public health research and practice: Although many study participants expressed trust in political measures at the beginning, this pro-portion declined during the course of the pandemic. The decrease in trust observed in our study might reflect families’ dissatisfaction or lack of understanding in the face of constantly changing restriction measures and relaxations. For example, the rules for closure or opening of nurseries have changed continuously, impeding families’ organization of everyday life. The public debate about adequate safety measures in childcare facilities was marked by controversies, and the actual implementation of safety measures differed be-tween different regions of Germany. However, it is essential for the management of the pandemic that people understand why measures are implemented. Policy measures should be comprehensible and consistent and better communication about why a specific measure is implemented could eventually contribute to people’s acceptance and trust. A further implication can be derived from our finding of variability of worries and trust over time and between regions. People in severely affected regions expressed stronger concerns, but they might also be more prone to implement safety measures. Policy measures and the associated communication strategy should acknowledge regional differences and be adapted to the local situation.”

Reviewer 2 Report

The authors analyse worries related to the Covid-19 pandemic and trust in anti-pandemic policy in families with children aged 1.5 to 5.9 years in two regions in Germany during the first and second wave of the Covid-19 pandemic using two longitudinal childhood cohorts in Germany. 720 parents (507 from Regensburg, Bavaria and 213 from Leipzig, Saxony) participated. Overall, most often participants indicated worries about possible impact of the pandemic on the economy; in contrast, worries about their own financial situation were comparably infrequent. Most reported worries increased from the first to the second Covid-19 wave. Trust in anti-pandemic policy measures was rated as moderately high, however significantly decreased from wave one to wave two. Authors conclude that to acknowledge regional differences might help families to cope with the burden of the Covid-19 pandemic. 

Major concerns:

  1. Study participants: as former studies identified families with low SES, migrational background or other psychosocial problems especially at risk for worries and problems related to the Covid-19 pandemic and anti-pandemic policy measures, it seems difficult to conduct the present study in two selected samples of families with high socio-economic status, high rates of employment and a low rate of migrational background. Other risk factors for worries regarding Covid-19 and the respective measures, like limited living space, parental mental health problems or loss of employment were not assessed or analysed. Hence, the results of the present study are applicable for very selected families and especially do not represente the worries of families at risk.
  2. Questionnaire: The questionnaire was developed by the authors to assess worries regarding Covid-19. It seems not thouroughly suited to describe the situation of the families comprehensively e.g. Question 5 "Change in working situation": The possible answers refer to home office, more or less or not. They do not cover the scope of possible developments like e.g. loss of work because of Covid-19. e.g. Question 9 (and consecutive) asks for worries regarding dangers of corona and the limitations of countermeasures. It is not possible to differentiate between worries regarding corona and worries regarding the countermeasures, though it has to be assumed that these worries might be quite diverging.
  3. As no standardized instruments were used, comparability with pre-pandemic data or other samples is extremely limited. 
  4. Altogether it seems that the assessment of risk factors as well as protective factors regarding worries of families related to the Covid-19 pandemic is quite restricted, hence transfereability of results to families in Germany is limited. 

Minor concerns: 

  1. Introduction: line55/56 the sentence seems to be incomplete.
  2. Table 1: migration background Leipzig W1: Percentages do not sum up to 100%. 
  3. Discussion: I propose to discuss the results clearly in light of the above mentioned limitations.
  4. Limitations: I propose to describe the limitations more comprehensively. 
  5. Conclusions should be adjusted regarding the above mentioned limitations. 

Author Response

RESPONSE

We would like to thank the reviewers and the editors for their suggestions and constructive feedback to our manuscript. Based on these, we have prepared a revised version of our manuscript.

Reviewer 2:

Comments and Suggestions for Authors

The authors analyse worries related to the Covid-19 pandemic and trust in anti-pandemic policy in families with children aged 1.5 to 5.9 years in two regions in Germany during the first and second wave of the Covid-19 pandemic using two longitudinal childhood cohorts in Germany. 720 parents (507 from Regensburg, Bavaria and 213 from Leipzig, Saxony) participated. Overall, most often participants indicated worries about possible impact of the pandemic on the economy; in contrast, worries about their own financial situation were comparably infrequent. Most reported worries increased from the first to the second Covid-19 wave. Trust in anti-pandemic policy measures was rated as moderately high, however significantly decreased from wave one to wave two. Authors conclude that to acknowledge regional differences might help families to cope with the burden of the Covid-19 pandemic. 

Major concerns:

  1. Study participants: as former studies identified families with low SES, migrational background or other psychosocial problems especially at risk for worries and problems related to the Covid-19 pandemic and anti-pandemic policy measures, it seems difficult to conduct the present study in two selected samples of families with high socio-economic status, high rates of employment and a low rate of migrational background. Other risk factors for worries regarding Covid-19 and the respective measures, like limited living space, parental mental health problems or loss of employment were not assessed or analysed. Hence, the results of the present study are applicable for very selected families and especially do not represente the worries of families at risk.

Response: We agree with the reviewer that selection bias limits the generalizability of our findings and that particularly families at risk for experiencing severe problems during the COVID-19 pandemic are not well represented in our study. Unfortunately, this applies to many population-based studies. We still believe that our study yielded important findings. We showed that worries are prevalent in many families and that the extent of worries increased during the course of the pandemic. However, it is probable that the situation is worse in families from poorer socio-economic background. We already discussed the selection bias and its possible consequences for the interpretation of our findings in the limitations section. In the revised version of the manuscript we have extended this line of reasoning even further.

  1. Questionnaire: The questionnaire was developed by the authors to assess worries regarding Covid-19. It seems not thouroughly suited to describe the situation of the families comprehensively e.g. Question 5 "Change in working situation": The possible answers refer to home office, more or less or not. They do not cover the scope of possible developments like e.g. loss of work because of Covid-19. e.g. Question 9 (and consecutive) asks for worries regarding dangers of corona and the limitations of countermeasures. It is not possible to differentiate between worries regarding corona and worries regarding the countermeasures, though it has to be assumed that these worries might be quite diverging.

Response: The selection of questions for the online survey reflected the topics that were discussed regarding the situation of families at the beginning of the pandemic. At this time, one of the main issues was how families (in particular women) managed to continue working without child care and how working from home was feasible with young children. In Germany, unemployment due to the COVID-19 pandemic was rare, at least during the first months, since employers were compensated by “Kurzarbeit”, a financial scheme allowing for avoiding unemployment. Nevertheless, job loss is an important pandemic-related concern we did not capture in our study, and we included this in the limitation section.

We agree with the reviewer that the items we used to assess parental worries cannot differentiate whether the worries are caused by the disease or by anti-pandemic measures. We now refer to this limitation in the discussion section.

Changes in the manuscript:
Discussion: “Additional limitations concern the selection of questions for the survey. Some important pandemic-related worries such as fear of losing the job were not captured. Regarding worries about oneself, family, friends, hometown, country, and the world, it is not clear whether the worries are related to COVID-19 or to anti-pandemic measures.”

  1. As no standardized instruments were used, comparability with pre-pandemic data or other samples is extremely limited. 
    Response: Yes, comparability with pre-pandemic data is extremely limited. The questions we used refer specifically to the situation in the COVID-19 pandemic. However, our data can be compared with some other studies which were also conducted during the COVID-19 pandemic and which used similar measurement instruments. Some of these studies are mentioned in the discussion section.

Changes in the manuscript:
Discussion “Finally, we could not use an established  validated measurement instrument but the questionnaire was developed by the authors. Therefore, comparisons with other studies are only possible to a very limited extent.”

  1. Altogether it seems that the assessment of risk factors as well as protective factors regarding worries of families related to the Covid-19 pandemic is quite restricted, hence transfereability of results to families in Germany is limited. 
    Response: An in-depth assessment of risk factors as well as of protective factors was beyond the scope of our study. Our main aim was to investigate what families worry about and how worries develop over time and between regions. However, some variables which could be indicative for families’ better or poorer management of the pandemic were investigated. We analysed data on migration background and education and considered them in the multivariable regression models. We now have expanded the limitations section and refer to the situation of families at risk for suffering from pandemic related worries.

Changes in the manuscript:
Discussion: “Due to this selection bias generalizability of our findings is limited.  People who were more concerned about the pandemic or who considered the pandemic a topic relevant to be investigated might have been more prone to participate in the present surveys. On the other hand, we may underestimate the impact of the pandemic on families who are underprivileged, and who are more severely affected by the pandemic, e.g., because their children attend more frequently child care centers with higher infection rates[37] or because they are single parents and, therefore, experience more care-related worries [38].-2

Minor concerns: 

  1. Introduction: line55/56 the sentence seems to be incomplete.

Response: We have corrected this sentence.

  1. Table 1: migration background Leipzig W1: Percentages do not sum up to 100%. 

Response: Thank you for catching this mistake. There were 52 missings (25%), not 42 (20%).

  1. Discussion: I propose to discuss the results clearly in light of the above mentioned limitations.

Response: The main limitations of our study are the selection bias and the underrepresentation of families with low socio-economic status. We refer to these issues at some points in the discussion section:

Discussion:
Low worries about own financial situation: “One reason for this discrepancy might be the low proportion of people with migration background and the large proportion of people with higher education, indicating a high socio-economic status of the study participants. In higher social strata, people are more likely to have stable jobs and financial reserves and, therefore, suffer less from the pandemic-related economic crisis [18]. Other studies found that particularly socially disadvantaged families were affected by lockdown measures [19,20] .”

Discussion:
Low worries about oneself: “The low level of worries about themselves might be explained by the higher social status in the present sample, as people from higher social classes are at lower risk for health risks at work [18].”

Discussion: “A limitation of our study is that the study population consisted mostly of well-educated parents as we drew them from our ongoing pediatric childhood cohort studies. As with most studies relying on active participation [35,36], families with low education, low in-come, and migration background were underrepresented. Due to this selection bias gener-alizability of our findings is limited., w On the other hand, pPeople who were more con-cerned about the pandemic or who considered the pandemic a topic relevant to be inves-tigated might have been more prone to participate in the present surveys. On the other hand, we may underestimate the impact of the pandemic on families who are underprivi-leged, and who are more severely affected by the pandemic, e.g., because their children at-tend more frequently child care centers with higher infection rates [37] or because they are single parents and, therefore, experience more care-related worries [38].”

  1. Limitations: I propose to describe the limitations more comprehensively. 

Response: According to your suggestions, we have revised the limitation section. Limitations of the study are now presented more comprehensively. We address particularly the problems arising from selection bias and the questionnaires which were used.

  1. Conclusions should be adjusted regarding the above mentioned limitations. 

Response: We weakened the conclusion that we drew from our findings and are stating now that our findings do not necessarily apply to the general population but to samples of predominantly well-situated families. The conclusion now reads as follows:

Changes in the manuscript:
Conclusion“ Investigating two large samples of predominantly well-situated families in Germany we found that  COVID-19-related worries and their trust in anti-pandemic policy measures changed during the course of the pandemic and differed remarkably between the study regions.”

Round 2

Reviewer 1 Report

I thank the authors for providing a revised version of their manuscript.

Some comments:

Please check the format of the main text and errors such as .) in the keyword list. Usually, 95% CIs are also reported with ORs in the text.

Please remove the brakets in table 2 when reporting CIs.

Author Response

I thank the authors for providing a revised version of their manuscript.

Response: We thank the reviewer for the helpful comments. 

Some comments:

  1. Please check the format of the main text and errors such as .) in the keyword list. Usually, 95% CIs are also reported with ORs in the text.

Response: Thank you for this comment. We removed the “)” from the keywords list. We added the 95% CI to the text. Regarding formal/other small changes, we changed the following:

  • We removed bold case for all references in the text
  • We checked that the text was formatted as requested in the template
  • We corrected formatting of Table 1 (there was an error in the last lines)
  • We corrected typing errors
  • We removed parts of the caption in Table 2 and included them in the Statistical analysis part
  • In the Discussion, we changed wave 2 to wave 1 at one point, as wave 2 was not correct
  1. Please remove the brakets in table 2 when reporting CIs.

Response: We removed the brackets in Table 2 when reporting OR.
